# Manipulating the Microbiome: An Alternative Treatment for Bile Acid Diarrhoea

**Evette B. M. Hillman** [1,2,*] **, Sjoerd Rijpkema** [1] **, Danielle Carson** [1] **, Ramesh P. Arasaradnam** [3,4] **, Elizabeth M. H. Wellington** [2] **and Gregory C. A. Amos** [1,*]

1   Division of Bacteriology, National Institute for Biological Standards and Control, South Mimms, Potters Bar, Hertfordshire EN6 3QG, UK; Sjoerd.Rijpkema@nibsc.org (S.R.); Danielle.Carson@nibsc.org (D.C.)
2   School of Life Sciences, The University of Warwick, Coventry CV4 7AL, UK; E.M.H.Wellington@warwick.ac.uk
3   Warwick Medical School, The University of Warwick, Coventry CV4 7AL, UK; R.Arasaradnam@warwick.ac.uk
4   University Hospital Coventry & Warwickshire, Coventry CV2 2DX, UK
*   Correspondence: Evette.Hillman@nibsc.org (E.B.M.H.); Gregory.Amos@nibsc.org (G.C.A.A.); Tel.: +44-(0)1707-641538 (E.B.M.H.); +44-(0)1707-641431 (G.C.A.A.)

**Abstract:** Bile acid diarrhoea (BAD) is a widespread gastrointestinal disease that is often misdiagnosed as irritable bowel syndrome and is estimated to affect 1% of the United Kingdom (UK) population alone. BAD is associated with excessive bile acid synthesis secondary to a gastrointestinal or idiopathic disorder (also known as primary BAD). Current licensed treatment in the UK has undesirable effects and has been the same since BAD was first discovered in the 1960s. Bacteria are essential in transforming primary bile acids into secondary bile acids. The profile of an individual's bile acid pool is central in bile acid homeostasis as bile acids regulate their own synthesis. Therefore, microbiome dysbiosis incurred through changes in diet, stress levels and the introduction of antibiotics may contribute to or be the cause of primary BAD. This literature review focuses on primary BAD, providing an overview of bile acid metabolism, the role of the human gut microbiome in BAD and the potential options for therapeutic intervention in primary BAD through manipulation of the microbiome.

**Keywords:** bile acid diarrhoea; irritable bowel syndrome; microbiome; bile salt hydrolase; bile acid transformation; faecal microbiota transplantation

## 1. Introduction

Bile acid diarrhoea (BAD) (previously referred to as bile acid malabsorption or bile salt malabsorption) is a condition that predominantly presents as chronic watery diarrhoea as well as bloating and abdominal pain [1]. A recent survey by Walters et al. (2020) observed that 86% of participants experienced frequent bowel movements, often more than 6 times a day [2]. In a different survey, Bannaga et al. (2017) reported that many BAD patients often experience tiredness, low energy levels and a lack of concentration [3]. Sufferers reported the negative impact BAD had on their social life and ability to work (which could lead to the loss of employment), and over 90% also felt a negative impact on their mental health. Although this condition is not life-threatening, the survey revealed that depression, isolation, helplessness and low self-esteem are very common among patients living with BAD, often leaving individuals housebound [3]. While the exact figures are not known, it is estimated that up to 1% of the population in the United Kingdom (UK) have primary BAD [4]. If non-primary BAD patients are included, this potentially results in an excess of 700,000 individuals in the UK alone [5]. Smith et al. (2000) estimated that over one-third of patients diagnosed with irritable bowel syndrome (IBS) actually suffer from BAD [6]. Consequently, BAD could have a wide-reaching economic impact due to absence from work through sickness and the cost of continued care.

BAD has several different causes that fall under one of three types [7]:

- Type I, when the ileum is damaged due to inflammation or surgical removal.
- Type II is idiopathic and is also known as "primary BAD".
- Type III, results from another disease or condition (such as gallbladder removal).

In 2009, Walters et al. proposed that primary BAD (type II) was not a result of reduced or impaired absorption but in fact resulted from unregulated bile acid synthesis [8]. Studies demonstrated that patients with BAD had significantly higher levels of $7\alpha$-hydroxy-4-cholesten-3-one ((C4), a bile acid precursor) and significantly lower levels of fibroblast growth factor 19 ((FGF19), a hormone that down regulates bile acid synthesis) compared with healthy individuals [8]. Similarly, the authors also showed that patients with secondary BAD (type I and III) had lower levels of FGF19 in their blood serum compared with that in healthy subjects [8–10]. This excessive bile acid synthesis results in the insufficient absorption of bile acids in the ileum, causing the increased transit of bile acids into the colon. High levels of bile acid in the colon trigger a rise in water secretion to combat irritation, which prevents stool from properly forming in the lumen, leading to diarrhoea. Several studies have reported the importance of bile acids in a number of gut related diseases [11]. Despite these observations, the cause of low serum levels of FGF19 and the consequent excess bile acid synthesis remains widely unknown.

The selenium-75-homocholic acid taurine (SeHCAT) scan (a nuclear medicine test) is considered the gold standard for diagnosing BAD, as it has a high sensitivity and specificity [12]. The parameters for diagnosing BAD are: less than 5% bile acid retention indicates severe BAD; between 5 to 10% bile acid retention indicates moderate BAD; and between 10 to 15% bile acid retention indicates mild BAD [4]. Despite the SeHCAT scan giving the highest diagnostic accuracy for BAD, it is not widely used or licensed in many countries (including the United States of America (USA)). In the absence of a SeHCAT scan, doctors often prescribe bile acid sequestrant medication as the response to treatment can provide a diagnosis. Bile acid sequestrants bind to excess bile acids with a high affinity preventing irritation in the large intestine. Frequently treatment for BAD is not administrated because it is often misdiagnosed as diarrhoea predominant IBS (IBS-D) due to the similarity in clinical presentation of both diseases. Indeed, Bannaga et al. (2017) reported that 44% of respondents had experienced symptoms for more than five years before diagnosis, with a range of between one and thirty years [3]. IBS is a multifactorial spectrum disease, which can be triggered by both environmental and genetic factors, although the exact cause of IBS is widely unknown [13]. There is no accepted diagnostic test for IBS; however, as mentioned, there is one for identifying BAD. IBS is not to be confused with inflammatory bowel disease (IBD), which presents as chronic inflammation on the GI tract.

Current treatment for BAD includes diet changes and medication; however, these only treat the symptoms and do not treat the cause of the disease. Physicians advise a diet with a reduced fat intake of 40 g of fat per day [14]. To date, there have been no randomised trials that have evaluated different dietary regimes with BAD sufferers. Instead, observational studies seem to suggest that low fat diets are beneficial. Low fat diets are thought to result in reduced bile acid synthesis and could possibly alter the microbiome, all leading to less severe symptoms in some cases [15–17]. Current treatment guidelines for BAD have not significantly improved since the 1960s, and sequestrants can have adverse effects such as constipation, bloating, nausea and abdominal pain (especially when combined with other medications) [18,19]. The pathophysiology of BAD is not fully understood, and, to date, a clinically effective cure remains elusive. This review will cover the biochemical structure of bile acids, the regulation of bile acid synthesis and the presentation of recent evidence towards the manipulation of the microbiome as a possible alternative treatment for primary BAD.

## 2. Structure and Function of Bile Acids

Bile acids are cyclo-pentane phenanthrene sterol molecules and are the main component of bile. The chemical composition of bile varies between individuals but usually consists of 61% bile acids, 12% fatty acids, 9% cholesterol, 7% proteins and small amounts of bilirubin [20]. Bile acids are amphiphilic molecules that possess a hydrophobic aliphatic steroid nucleus with 24 carbon atoms, which are attached to hydrophilic hydroxyl groups and an acidic side chain. The steroidal core of bile acids consists of a saturated four ring skeleton, consisting of three six-membered rings (A, B, and C) and one five-membered ring (D) (Figure 1).

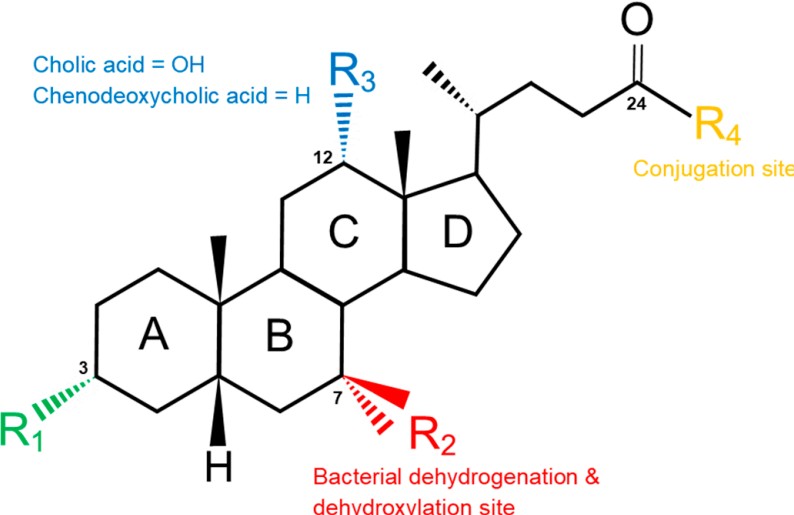

| Name | $R_1$ | $R_2$ | $R_3$ | $R_4$ |
|---|---|---|---|---|
| Cholic Acid | OH (α) | OH (α) | OH | OH |
| Chenodeoxycholic Acid | OH (α) | OH (α) | H | OH |
| Deoxycholic Acid | OH (α) | H | OH | OH |
| Lithocholic Acid | OH (α) | H | H | OH |
| Ursodeoxycholic Acid | OH (α) | OH (β) | H | OH |
| Glycocholic Acid | OH (α) | OH (α) | OH | $NHCH_2COO^-$ |
| Glycochenodeoxycholic Acid | OH (α) | OH (α) | H | $NHCH_2COO^-$ |
| Glycodeoxycholic acid | OH (α) | H | OH | $NHCH_2COO^-$ |
| Glycolithocholic acid | OH (α) | H | H | $NHCH_2COO^-$ |
| Gycoursodeoxycholic Acid | OH (α) | OH (β) | H | $NHCH_2COO^-$ |
| Taurocholic Acid | OH (α) | OH (α) | OH | $NHCH_2CH_2SO_3^-$ |
| Taurochenodeoxycholic Acid | OH (α) | OH (α) | H | $NHCH_2CH_2SO_3^-$ |
| Taurodeoxycholic acid | OH (α) | H | OH | $NHCH_2CH_2SO_3^-$ |
| Taurolithocholic acid | OH (α) | H | H | $NHCH_2CH_2SO_3^-$ |
| Tauroursodeoxycholic Acid | OH (α) | OH (β) | H | $NHCH_2CH_2SO_3^-$ |
| 7-keto-deoxycholic acid | OH (α) | O | OH | OH |
| 7-keto-lithocholic acid | OH (α) | O | H | OH |
| 12-keto-lithocholic acid | OH (α) | H | O | OH |
| Isolithocholic acid | OH (β) | H | H | OH |

**Figure 1.** The schematic molecular structure of common bile acids found in humans annotated to show its relationship to other bile acids. α (represented by a dashed line) indicates a steric downwards orientation and β (represented by a solid line) indicates a steric upwards orientation.

The covalently bonded hydroxyl groups are in one of either two configurations: α, a downwards orientation; or β, an upwards orientation. Almost all bile acids have an α-hydroxyl group at position C-3, which is derived from the parent molecule cholesterol,

which has the 3-hydroxyl group in the β orientation. There are two types of bile acids: primary and secondary bile acid. Primary bile acids in humans can be further divided between 3α, 7α, 12α-trihydroxy-5β-cholan-24-oic acid (or cholic) acid (CA) and 3α, 7α-dihydroxy-5β-cholan-24-oic acid (or chenodeoxycholic) acid (CDCA). These bile acids differ only at position C-12, wherein CA possess a hydroxyl (OH) group whereas CDCA does not. Primary bile acids are often conjugated via N-acyl amidation with an amino acid, as 95% of these conjugated bile acids are bound to either glycine or taurine at position C-24 (see Figure 1) [21]. Of this 95%, approximately 75% are conjugated to glycine and 25% are conjugated to taurine. Unconjugated bile acids have a pKa value range of 5–6. The addition of an amino acid lowers the pKa value of glycine and taurine conjugated bile acids from 5–6 to 4–5 and 1–2, respectively. At the physiological pH of 7.0–7.7, bile consists predominantly of conjugated bile acids, which are ionised and are alternatively known as bile salts or strong bile acids [22]. Bile salts are more water-soluble and are able to fulfil one of their physiologic functions of solubilising dietary lipids due to their increased amphipathic nature. The added solubility prevents the passive reabsorption of conjugated bile acid in the small intestine, causing a reliance on active reabsorption via the apical sodium–bile acid transporter (ASBT) in the ileum.

Bile acids have a number of functions in the gastrointestinal (GI) tract: conjugated bile acids at the right concentration form micelles, which function as biological detergents through the emulsification and solubilisation of fats in the small intestine [23]; they are essential for the absorption of dietary lipids (including lipoproteins and fat-soluble vitamins such as vitamin D); they act as a signalling molecules that regulate their own expression; they play a key role in immune homeostasis and the metabolism of glucose [24,25]. Due to their amphipathic detergent nature, bile acids are toxic to microbes and act as antimicrobial agents through membrane damage, thus helping to maintain homeostasis between bile acid-tolerant and bile acid-sensitive bacteria in the GI tract [26]. Even at sub-micellar concentrations, bile acids can alter bacterial membrane lipid composition [27]. In one of the few studies that has investigated the bile acid profile in SeHCAT diagnosed BAD patients, Sagar et al. (2020) reported an increase in the concentrations of bile acids (particularly CDCA) in the faeces and serum of BAD patients compared to those in IBS-D patients [28].

## 3. The Enterohepatic Circulation and Primary Bile Acid Synthesis

Bile acids are synthesised primarily by hepatocytes [29,30]. On average, the gallbladder secretes 12–18 g of bile acids per day into the intestine, typically following meals. However, in healthy individuals there are approximately 4–6 g of bile acids present in the intestine at any given time due to the enterohepatic circulation (EHC) of bile acids [31]. Approximately 95% of all intestinal bile acids are reabsorbed by passive diffusion in the intestine or, in the case of conjugated bile acids, by active transport. Following reabsorption from the intestine, bile acids are recycled by the liver, with any unabsorbed bile acids remaining in the intestine (~400–800 mg, or 5%) and passing to the colon for excretion [32].

Cholesterol is an obligatory precursor of bile acids (Figure 2). The conversion involves several enzymatic processes; hepatocytes express all 17 enzymes necessary for modifying the steroid core, removing the side chain and conjugating with an amino acid, resulting in the synthesis of primary conjugated bile acids [33]. This occurs through one of two pathways: the classic (neutral) pathway (which occurs ~90% of the time in humans and ~75% of the time in mice) or the alternative (acidic) pathway (occurs ~10% of the time in humans and ~25% of the time in mice) [33–35]. The principal rate-limiting enzyme (cholesterol-7α-hydroxylase of the cytochrome P450 family 7 subfamily A member 1 (CYP7A1)) is required for bile acid biosynthesis in the classic pathway. CYP7A1 catalyses the hydroxylation of cholesterol at C-7, creating 7α-hydroxycholesterol [36]. There are several intermediates in the production of bile acids, including C4, which is sometimes used as a biomarker in diagnosing BAD [37,38]. Sterol-27-hydroxylase (CYP27A1, cytochrome P450 family 27 subfamily A member 1) is a mitochondrial enzyme that is responsible for facilitating

bile acid synthesis in the alternative pathway. However, unlike in the classic pathway, the hydroxylation of cholesterol takes place at C-27 [34,35].

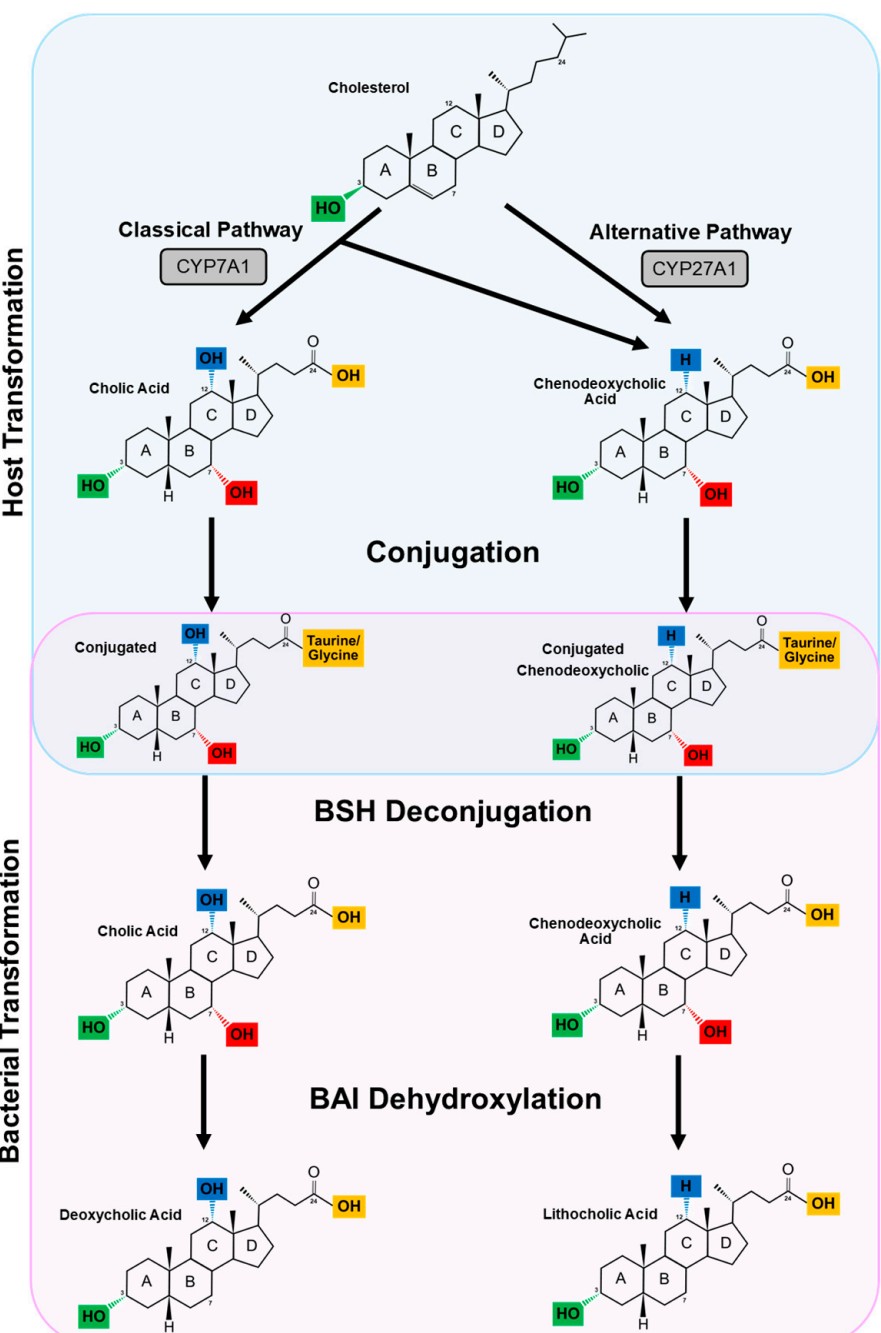

**Figure 2.** Host and bacterial primary and secondary bile acid metabolisms. Host transformation from cholesterol by enzymes (boxed) cytochrome P450 family 7 subfamily A member 1 (CYP7A1), in the classical pathway and CYP27A1 in the alternative pathway produces cholic acid and chenodeoxycholic acid, which are conjugated with taurine or glycine. Bacterial transformation often requires deconjugation via bile salt hydrolases (BSH), which exposes the bile acid to a range of further modifications. For simplicity, only deoxycholic acid and lithocholic acid are shown here, following dehydroxylation by enzymes encoded in the bile acid-inducible (bai) operon.

## 4. Secondary Bile Acids

In both the small intestine and in the colon (particularly in the colon), unabsorbed bile acids are deconjugated and further transformed into secondary bile acids by bacteria

(Figure 2). Bacterial enzymes can manipulate bile acids at four main sites on the sterol core: C-3, C-7, C-12 and C-24 (see Figures 1 and 2). C-24 is the site for deconjugation (the removal of the amino acid), C-7 is the site for dehydroxylation (the removal of hydroxyl group) or dehydrogenation (the removal of a hydrogen atom) and C-3, C-7 and C-12 are the sites for dehydrogenation. Bacterial dehydrogenation is a reversible process, as dehydrogenases catalyse the oxidative reactions in both directions.

### 4.1. Deconjugation

The conversion of conjugated bile acids by bacteria in the GI tract requires the bile salt hydrolase (BSH, EC 3.5.1.24) enzyme, a member of the choloylglycine hydrolase family that catalyse the hydrolysis of glycine or taurine from the C-24 position of the sterol core, resulting in deconjugated bile acids [39,40]. The active site of BSHs share several conserved amino acids (such as cysteine 2 (Cys2)) [41]. However, bacterial BSHs have widely different catalytic efficiencies as well as substrate specificities. In addition, bacterial BSHs differ in: genetic properties (gene organisation and regulation); physical properties (composition, size and weight of each subunit); and kinetic properties ($V_{max}$, optimum pH's and optimum temperatures). BSHs have been observed in numerous microbial genera and are predominantly described among Gram-positive bacteria, including *Lactobacillus* [42] and *Bifidobacterium* [43] which are often used as probiotics, as well as *Enterococcus* [44] and *Clostridium* spp. [45,46]. Furthermore, BSHs have been identified in Gram-negative bacteria such as and Bacteroides [45]. Recently, Song et al. (2019) elucidated the abundance of bacterial genes that encode BSHs among 11 human populations from geographically diverse origins [47]. After screening the Human Microbiome Project (HMP) database, they reported a total of 26% of bacteria strains carrying genes that encode BSHs [47]. These genes were widely distributed across 117 bacterial genera of the human microbiota, with the majority (71%) observed in *Bacteroides*, *Blautia*, *Eubacterium*, *Clostridium* and *Roseburia* [47]. How enzyme activity and specificity differ across strains is yet to be characterised; however, only *Lactobacillus* possessed the variant of BSH (BSH-T3) which has the highest known enzymatic activity amongst BSHs [47]. Studies suggest that BSHs have a preference for glycine conjugated bile acids, with the accumulation of taurine conjugated bile acids observed following the reduction of BSH-expressing bacteria in humans [48,49]. There are a number of benefits that BSHs confer to the bacterial host, including providing a source of nutrients [43,50,51]. Previously, it was proposed that deconjugation could protect bacteria from bile salt toxicity. However, recent studies have demonstrated unconjugated bile acids possess more potent antibacterial activity on *Staphylococcus aureus*, *Ruminococcus bromii*, *Bacteriodes fragilis*, *Lactobacillus paracasei*, *Bifidobacterium longum*, and *Clostridium scindens* than conjugated bile acids [52,53]. This is potentially due to unconjugated bile acids passively crossing membranes, resulting in intra-cellular toxicity while conjugated bile acids require a transporter and are thus more limited in their activity.

### 4.2. Dehydroxylation and Dehydrogenation

Bacterial genes that regulate the dehydroxylation and dehydrogenation of bile acids are encoded in the bile acid-inducible (*bai*) operon [41]. The first step of dehydroxylation requires baiB that encodes bile acid CoA ligase, resulting in a bile acid-CoA conjugate. Other key genes include baiE (which encodes 7α-de-hydratase) and baiA (which encodes 3α-hydroxysteroid dehydrogenases), resulting in the removal of a 7α hydroxyl group and a 3α hydrogen atom, respectively.

#### 4.2.1. Dehydroxylation

Only a limited number of bacterial species possess the bai operon required for the 7α-dehydroxylation of bile acid [54]. In the human intestine, 7α-dehydroxylation results in secondary bile acids: deoxycholic acid ((DCA); 3α,12α-dihydroxy-5β-cholan-24-oic acid) and lithocholic acid ((LCA); 3α-hydroxy-5β-cholan-24-oic acid) from CA and CDCA, respectively. DCA and LCA are the most abundant bile acids found in human faeces,

making up 34% and 29%, respectively [41]. Therefore, 7α-dehydroxylation is very quantitatively important in the bio-transformation of bile acid in the human colon. Genes encoding for dehydroxylation are most commonly observed in *Clostridium* spp. [55]. Recently, *Eubacterium* spp. have also been characterised to rapidly produce DCA from CDCA, and metagenomic approaches have observed the presence of 7α-dehydroxylation in *Ruminococcaceae*, *Lachnospiraceae*, and *Peptostreptococcaceae* [56,57]. Unlike bile acid C-3, C-7 and C-12 dehydrogenation, 7α/β-dehydroxylation is restricted to free, unconjugated bile acids. Therefore, the removal of the glycine or taurine via a BSH is an essential prerequisite for 7α/β-dehydroxylation by the intestinal microbiota [58,59].

### 4.2.2. Dehydrogenation

Dehydrogenation by hydroxysteroid dehydrogenases (HSDHs) leads to the oxidation, reduction and, often, epimerisation of the C-3-, C-7- and C-12-hydroxy groups of bile acids by 3α-HSDHs, 7α/β-HSDHs and 12α/β-HSDHs, respectively. Epimerisation is the reversible change in stereochemistry from an α → β or β → α configuration with the generation of a stable carbonyl group bile acid intermediate. 7-keto-lithocholic acid (7-keto-LCA) is the intermediate for the oxidation and reduction at C-7 of epimers CDCA and ursodeoxycholic acid (UDCA). UDCA epimerisation requires the combined effort of two position-specific, stereochemically distinct HSDHs. Such HSDHs can originate from the same or different species demonstrating the importance of synergy in the microbiome. For example, the presence of both 7α- and 7β-HSDH in *C. absonum* allows for epimerisation by a single bacterial species [60–62]. Meanwhile, epimerisation has been demonstrated in co-cultures of intestinal bacteria, with one possessing 7α-HSDH and the other 7β-HSDH [63]. To date, it has been difficult to quantify the degree of α/β-dehydroxylation that occurs in the gut due to the competing and reversible α/β-dehydrogenation of bile acids. Oxidation of the bile acid hydroxyl groups is believed to generate NADH, an important enzyme in making ATP that can be used by the bacterium [64].

3α-HSDHs have been detected in a number of *Clostridium* species, including *C. perfringens* [65], *C. scindens* [66] and *C. hiranonis* [67]. They have also been observed in other intestinal bacterial species, such as *Eubacterium lentum* [68] and *Peptostreptococcus productus* [69]. 3β-HSDH activity has been described in species of *Clostridium*, *Rumminococcus* and *Peptostreptococcus productus* [69–72]. Most 3α/β-HSDH genes described to date are expressed constitutively, though 3α/β-HSDHs from *C. scindens* and *C. hiranonis* are induced by CA and CDCA. [41].

7α/β-HSDHs are thought to be constitutively produced and are common among gut microbiota species [41]. Many intestinal clostridia express genes responsible for the production of both 7α- and 7β-HSDHs that epimerise the 7α/β-hydroxy group [62,73]. Several species with 7α/β-HSDH activity (such as *Ba. fragilis* [74], *C. sordellii* [75], *C. perfringens* [76] and *C. innocuum* [71]) also contain genes that encode for BSHs. The physiological significance of the bile acid 7α-hydroxy group epimerisation to gut bacteria is thought to decrease the toxicity of CDCA, as UDCA is less hydrophobic [77,78].

12α/β-HSDHs have been detected mainly among members of the genus *Clostridium*. 12α-HSDHs has been observed in *C. perfringens* [65], *C. leptum* [79], *Clostridium* group P [80] and *E. lentum* [68]. 12β-HSDHs have been reported in *C. tertium*, *C. difficile*, and *C. paraputrificum* [81]. Like 3 and 7α/β-HSDHs, most 12α/β-HSDHs are constitutively produced; however, the 12β-HSDH produced by *C. paraputrificum* is induced by 12-carboxyl-bile acid substrates [82].

### 4.3. H+ Bile Acid Transporter

7α/β-dehydroxylation activity and, therefore, the transformation of primary bile acids into the most commonly found secondary bile acids in human stool, requires the transport of free primary bile acids into the bacterial cell. This is carried out by the proton-dependent bile acid transporter coded for this purpose by baiG, which facilitates the transport of the unconjugated primary bile acids CA and CDCA but not of the secondary bile acids DCA

and LCA [83]. Bacterial species capable of primary bile acids 7α-dehydroxylation possess baiG [84].

## 5. Bile Acid Receptors and Downstream Targets

Bile acids can act as signalling molecules which bind to receptors that are involved in several vital signalling networks [24,25]. Bile acids are not only ligands for the farnesoid X receptor (FXR), which is the nuclear receptor superfamily 1 group H member 4 (NR1H4), but also other members of the nuclear receptor superfamily, such as the androstane receptor (NR1H3 or CAR), the pregnane X receptor (NR1H2 or PXR) and the vitamin D receptor (NR1H1 or VDR) [85]. In addition, secondary bile acids interact with a number of membrane receptors, including the G-protein-coupled bile acid receptor 1 (GP-BAR1; also known as TGR5). Of these receptors, FXR and TGR5 have been implicated as therapeutic targets to treat BAD.

### 5.1. Bile Acid Regulate Their Own Synthesis

The enterohepatic circulation of bile acids is tightly regulated. FGF19 is a hormone produced in the liver and ileum that suppresses the expression of the rate-limiting enzymes CYP7A1 and CYP27A1 that are central to bile acid synthesis (see Figure 3). FXR is a bile acid sensor, which is responsible for the regulation of FGF19 and is primarily expressed by the ileum, liver, adrenal glands and kidneys [86,87]. Bile acids (particularly the primary bile acids CDCA and CA) are natural agonists for FXR. Consequently, in healthy individuals, high concentrations of bile acid activate FXR, which upregulates FGF19, thus inhibiting bile acid production through the hepatic coreceptors FGF receptor 4 and β klotho [30]. A range of bile acids can bind to FXR, with differing binding affinities and ligand efficacies [88–90]. Wang et al. (1999) reported that UDCA did not activate FXR [90]; instead, Mueller et al. (2015) revealed that it inhibited FXR activation [91], in agreement with the in silico FXR binding data [92]. Recently, Zhao et al. (2020) observed a number of bile acids (glycochenodeoxycholic acid (GCDCA), glycoursodeoxycholic acid (GUDCA), glycocholic acid (GCA), UDCA and 7-ketodeoxycholic acid (7-KDCA)) isolated from human stool could efficiently antagonise CDCA-induced FXR activation, while taurine conjugated bile acids (specifically tauroursodeoxycholic acid (TUDCA) and taurine-conjugated-mix) can attenuate cellular FGF19 expression in vitro [93]. UDCA was one of the bile acids that antagonised CDCA-induced FXR activation and was shown to be significantly elevated in the stool of BAD and IBS-D patients with elevated total faecal bile acids [28,93]. Indeed, studies comparing the bile acid pool in vegan and omnivore diets report significantly more bile acids (including UDCA, glycine and taurine conjugated bile acids) in participants who consumed meat [94]. These observations indicate that diet may influence the bile acid pool. By elucidating the antagonist effects of bile acids, this data could explain the increase in bile acid synthesis in BAD patients. Furthermore, the antagonist effects of bile acids on FXR are well documented in mice, which express fibroblast growth factor 15 (FGF15), the murine homologue of FGF19 [95].

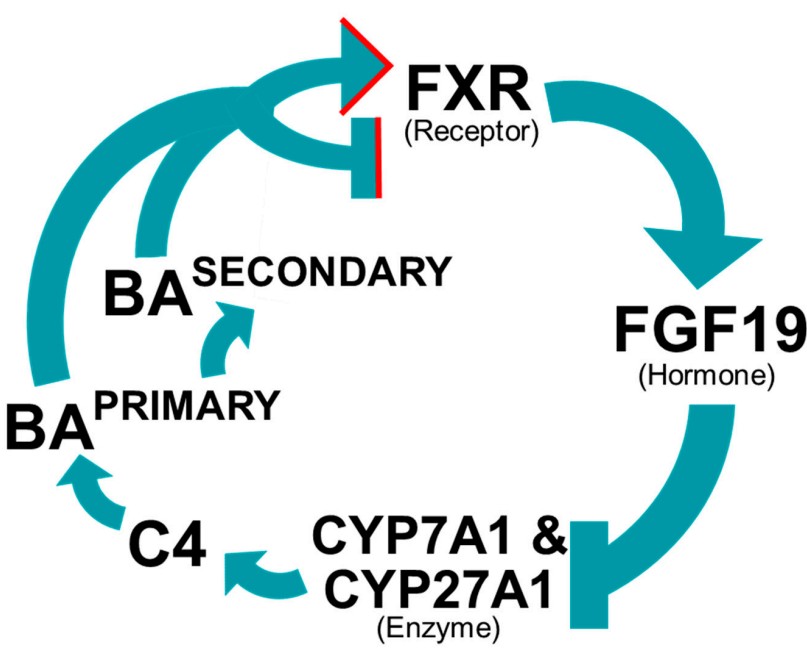

**Figure 3.** Bile acids regulate their own expression as primary and secondary bile acids bind as agonists and antagonists to the nuclear receptor FXR, which transcriptionally activates the hormone FGF19 that inhibits CYP7A1 thus inhibiting bile acid synthesis. FXR, farnesoid X receptor; FGF19, fibroblast growth factor 19; CPY7A1, cholesterol-7α-hydroxylase; CYP27A1, Sterol-27-hydroxylase; C4, 7α-hydroxy-4-cholesten-3-one; BA, bile acid.

### 5.2. Farnesoid X Receptor

FXR is a transcriptional activator which binds to form a hetero-dimeric complex with a retinoid X receptor (RXR) before binding to specific DNA response elements upstream of promoters [96]. The target DNA sequences are composed of two inverted repeats separated by one nucleotide. Most ligands that bind to RXR differ from those of FXR; for example, the natural agonist 9-cis-retinoic acid [97]. Interestingly, the ligands for both receptors can activate the transcriptional hetero-dimer, thus regulating bile acid synthesis. One of several FXR target genes is the small heterodimer partner (shp). SHP is a nuclear receptor that lacks a DNA binding domain; instead, it forms a hetero-dimer with both liver receptor homolog 1 and liver X receptor α. This complex, like FGF19, can inhibit CYP7A1 expression, thus inhibiting the classical pathway of bile acid synthesis [98]. To prevent toxification in the liver caused by bile acid, FXR negatively regulates bile acid uptake by hepatocytes via sodium taurocholate co-transporting polypeptide (NTCP) [99]. In addition, FXR activation represses human ASBT in the intestinal tissue. This is important, as ASBT transports bile acid from the lumen into enterocytes and induces basolateral organic solute transporters (OST) α and β that transport bile acids from enterocytes into sera [100].

### 5.3. Diet1 Protein

Vergnes et al. (2013) demonstrated that protein Diet1 regulates the production of FGF15/19 in mouse and human intestinal enterocytes [101]. The authors also observed that Diet1-deficient mice had reduced Fgf15 mRNA and protein levels in ileum, as well as increased plasma levels of bile acids particularly free and taurine conjugated CA [101]. Diet1 does not regulate transcription of FGF15/19, as it lacks the required DNA binding domains. It is hypothesised to physically interact with the protein [101]. Mutations in diet1 lead to differences in FGF15/19 expression and, subsequently, could contribute to variations in bile acid metabolism that may play a role in the development of BAD.

*5.4. TGR5 Receptor*

Present in the plasma membrane, TGR5 is expressed in many tissues, such as the gallbladder, spleen, liver and intestines, including the ileum and the colon [102–104]. Both primary and secondary conjugated and unconjugated bile acids constitute natural ligands for TGR5 [103]. LCA are thought to be the most potent natural ligand followed by DCA [103,104]. Inhibitory motor neurons form part of the enteric nervous system (ENS) and are responsible for inducing relaxations of intestinal smooth muscle. Studies have demonstrated that the activation of TGR5 by secondary bile acid agonist DCA and LCA inhibits spontaneous, phasic contractions of isolated colonic muscle in vitro [105]. In the intestinal tract, the G-protein-coupled receptor TGR5 is predominantly expressed by enteric neurons and enterochromaffin cells. Within the myenteric plexus, TGR5 is localised in ~50% of all neurons, mainly in the majority of inhibitory motor neurons and descending interneurons, which express nitric oxide (NO) synthase. NO is an important neurotransmitter and is also an anti-microbial agent important in host defence against enteric infection in murines and humans [105]. Moreover, in mice gavaged with DCA, delays in both gastric emptying and transit through the small intestine were observed [105]. Alemi et al. (2013) revealed that the activation of TGR5 by bile acids stimulated the release of neurotransmitters 5-hydroxytryptamine and the calcitonin gene-related peptide, which are both involved in peristalsis and cause involuntary movements of muscles cells of the intestine [106]. Furthermore, tgr5 (-/-) mice did not experience bile acid activated peristalsis or neurotransmitter release and had a whole-gut transit time 1.4-fold slower compared to that of wildtype. Thus, TGR5 activation by bile acids results in the BAD-like symptoms of diarrhoea and abdominal pain.

## 6. Microbiome

The term microbiome was first coined in the 1988 book *Fungi in Biological Control Systems* [107] and is currently broadly recognised as "the entire community of microorganisms (bacteria, archaea, lower and higher eukaryotes, and viruses), their genomes and the surrounding environmental conditions" [108]. The metazoan body hosts several microbiomes; a medley of cross species physico-chemical and chemical interactions that have a marked effect on the balance between host health and disease [109,110]. The role of the microbiome in health is one of the most researched areas of the 21st century, due in part to advances in microbial detection [111]. The human microbiome is separated and named depending on the site of the body they occupy, with the most well studied being the intestinal, skin, oral and vaginal microbiomes. This review focuses on the gut microbiome. The human microbiota consists mostly of bacterial cells (around $3.8 \times 10^{13}$) which have a marginally higher order compared to the number of human cells (around $3.0 \times 10^{13}$) [112]. Therefore, the ratio is around 1:1 which is quite different to previous values which estimated the human microbiome contained 10 times as many cells as the host [113]. Notably, this highlights an issue of microbiome research which will benefit from standardisation of analytical techniques [114].

*6.1. Gut Microbiome*

The intestinal tract is home to 70–80% of the human body's immune cells, which are shaped by the microbiome [115,116]. Moreover, around 200–600 million neurons capable of acting independently from the sympathetic and parasympathetic nervous systems (known as ENS) are embedded in the lining of the human GI tract [117,118]. Along the intestinal tract, the microbiota varies according to the anatomical region and organ they colonise. This is due to variations in pH, $O_2$ concentration, digesta flow rates (also known as trans time), nutrients and substrate availability [119]. Given that colon harbours the densest number of microbes out of all the organs [112]; the gut microbiome is central in the development and function of physical and mental, health and disease.

Tierney et al. (2019) analysed 3500 faecal samples and detected the presence of 22 million microbial genes in the gut microbiome [120]. As the gut microbiota co-evolved

with the host over thousands of years, these genes often have mutual and added benefits. Microbial genes play a critical role in facilitating host metabolism and the regulation of host physiology (such as in the removal, synthesis, and absorption of many fundamental nutrients and metabolites, including bile acids, lipids, amino acids, vitamins and short-chain fatty acids). Commensal bacteria have also been shown to protect the host against pathogens by consuming available nutrients, and often produce bacteriocin to inhibiting pathogenic growth and subsequent epithelium colonisation. Deficiencies or dysbiosis in the gut microbiome has been implicated in a wide range of diseases such as inflammatory bowel disease, celiac disease, colorectal cancer and BAD [121]. Disease is thought to result from imbalance in the host–microbe interaction when there is either a depletion or an increase of microbiota-derived metabolites such as short-chain fatty acids or bile acids. These metabolites interact with the host signalling and immune system to contribute, exacerbate or cause disease.

*6.2. Microbiome and Bile Acid Profile in Mice*

By observing and altering the microbiome as well as the bile acid pool in mice, scientists have demonstrated that intestinal bacterial homeostasis is maintained by bile acids and vice versa [95,122–126]. Studies in germ free (GF) mice revealed that the bile acid pool consists almost exclusively of primary conjugated bile acids as compared to that of conventionally raised mice [95,122]. These results verify that, in the absence of bacteria, deconjugation and secondary bile acid formation is greatly perturbed. Bile acids α and β muricholic acids (αMCA and βMCA) are derived from CDCA and, unlike those in humans, primary bile acids are almost exclusively conjugated with taurine in mice. The antagonistic effects of primary bile acids, taurine conjugated αMCA (TαMCA), taurine conjugated βMCA (TβMCA), glycine conjugated βMCA (GβMCA) and UDCA have been observed for FXR [91,95,127]. As bile acid deconjugation requires bacteria, Sayin et al. (2013) reported an accumulation of TβMCA in GF mice, reduced FXR signalling and a subsequently increased bile acid synthesis [95]. Interestingly, changes in the microbial community structure of mice following a gavage with FXR antagonists TαMCA, TβMCA and GβMCA, have been observed [123,124]. These observations confirm the importance of the microbiome in shaping the bile acid pool and these infer that manipulating the microbiome can be used as a treatment for BAD.

*6.3. The Gut Microbiome and Bile Acid Diarrhoea*

There have been very few studies to date on the gut microbiota of BAD patients (who were confirmed by SeHCAT scan). Sagar et al. (2020) revealed that the gut microbiome of BAD patients had evidence of dysbiosis, with significantly reduced bacterial diversity compared with IBS-D [28]. Indeed, despite IBS-D being a distinct and separate disease, BAD is often misdiagnosed as IBS-D; thus, much of the presumed knowledge on the microbiome of BAD patients is inferred from IBS-D studies. This is problematic when trying to evaluate an evidence-based role for the microbiome in BAD and highlights the importance of accurate clinical metadata in microbiome studies. For IBS-D, Duboc et al. (2012) observed reduced concentrations of the probiotic *Bifidobacterium* and the anti-inflammatory commensal *C. leptum* bacteria in the faecal microbiota of IBS-D patients [128]. In addition, the authors observed an overall decrease in the abundance of bacteria possessing the bsh gene required for bile acid transformation [128]. Consistent with these findings, a recent study demonstrated that individuals suffering from severe IBS-D, with elevated levels of bile acids in their stool, possessed a significantly reduced abundance of bsh and an elevated abundance of genes encoded by the bai operon [93]. Taken together, this suggests a lower level of bile acid transformation by the gut microbiota for individuals with IBS-D and potentially BAD, as the authors of these papers did not rule out BAD. Interestingly, although functional metagenomic analysis of the microbiota of BAD patients has yet to be performed, metabolomic analysis revealed that BAD patients have a lower proportion of secondary bile acids as compared to IBS-D patients [28]. Elevated concentrations of bile acids are

linked to accelerated colonic transit time which could subsequently result in reduced time for bile acid transformation to secondary bile acids by gut bacteria, thus exacerbating BAD symptoms. Indeed, the gut microbiome and its subsequent transformation of bile acids is central to maintaining the balance between FXR and TGR5 receptor activation via the transformation of primary bile acids to secondary bile acids. Therefore, manipulating the gut microbiome could be a permanent treatment for BAD. Moreover, changes in the gut microbiome due to external factors such as stress, antibiotics or diet could be the primary cause of BAD, which would explain why treatment to date by bile sequestrants does not alleviate the underpinning cause of the disease.

### 7. Manipulating the Microbiome

The multifaceted gut microbiota is dominated by bacteria from 12 different phyla that predominantly belong to four major phyla (Firmicutes, Bacteroidetes, Actinobacteria, and Proteobacteria) [129–131]. With millions of genes and subsequent proteins performing beneficial and vital functions, it is clear why many scientists view the microbiome as a symbiotic organ. Diet, the environment, stress and drugs such as antibiotics can all disrupt homeostasis of the gut by disrupting the diversity of the microbiome [132–134]. Manipulating the microbiome can result in favourable changes in the structure and function of the intestinal microbiota and their metabolites. Indeed, microbiome dysbiosis and subsequent disease symptoms have been effectively treated by altering the microbiome [135,136]. The following therapies could be used to treat BAD as bacteria are responsible for transforming primary bile acids into secondary bile acids which have been shown to antagonise FXR activation and possibly increase bile acid synthesis, resulting in BAD symptoms [91–93].

To date, faecal microbiota transplantation (FMT) is the only licensed microbiome based therapeutic. FMT involves the transfer of stool from a healthy donor to the recipient and it is used to treat diarrhoea; this practice in different forms has been utilised for over 1700 years [137]. Clinical studies, particularly on patients with a *C. difficile* infection and inflammatory bowel disease, have demonstrated this to result in very high success rates [135,136,138]. Recently El-Salhy et al. (2020) published promising results on the efficacy of FMT in IBS patients [139]. A therapeutic gain was seen in 89.1% of donors that received 60 g FMT from a single super-donor. However, Lahtinen et al. (2020) concluded a single transplant via colonoscopy is not a recommended treatment for IBS. It is likely that FMT differs in its mode of action for different diseases. Consequently, FMT therapy lacks control of dose and duration of exposure to the microbiome or its constituents.

Following on from the success of FMT, live bio-therapeutic products (LBPs) are being trialled for the treatment of disease by modulating the microbiome [140]. "An LBP is a manufactured biological product that contains live organisms used for prevention, treatment, or cure of a disease or condition in humans" [140]. LBPs are sometimes termed next-generation probiotics, as they function in much the same way as classical probiotics but are rationally designed based on microbiome composition rather than containing specific probiotic strains which use specific food groups as substrates. Examples of bacterial species commonly observed in food-based probiotics include the *Lactobacillus* genus (*L. acidophilus*, *L. plantarum* and *L. paracasei*), the *Bifidobacterium* genus (*B. breve*, *B. longum* and *B. infantis*) and *Streptococcus thermophilus* [141,142].

A study employing sterile faecal filtrate transfer (FFT) showed interesting results by successfully treating recurrent *C. difficile* infection in all five patients, suggesting that live microbes are not required to restore healthy stool habits [143]. The authors of this study concluded that bacterial components, metabolites (such as bile acids) and/or bacteriophages are vital in a successful FMT [143]. Prebiotics are bacterial substrates (such as fructo-oligosaccharides and galacto-oligosaccharides) that can be used in parallel with probiotics and LBPs (synbiotics) or as a standalone strategy, and are another way of altering the microbiome.

## 8. Conclusions

A vital way in which intestinal bacterial homeostasis is conserved in the GI tract is through bile acids that limit the growth of bile acid-sensitive bacteria and promote the growth of bile acid-tolerant bacteria. Data generated from IBS-D studies indicate that there is strong evidence that changes in the microbiome as a result of stress, and a change in diet or the taking of antibiotics could be responsible for bacterial dysbiosis and, subsequently, type II BAD (see Figure 4) [137,144–146]. High levels of taurine conjugation in bile acids have been observed in meat and seafood, suggesting that taurine conjugation is potentially associated with meat consumption [41,147]. As taurine conjugated bile acids have shown antagonistic effects on FXR activation, this may be a cause of dysbiosis [93].

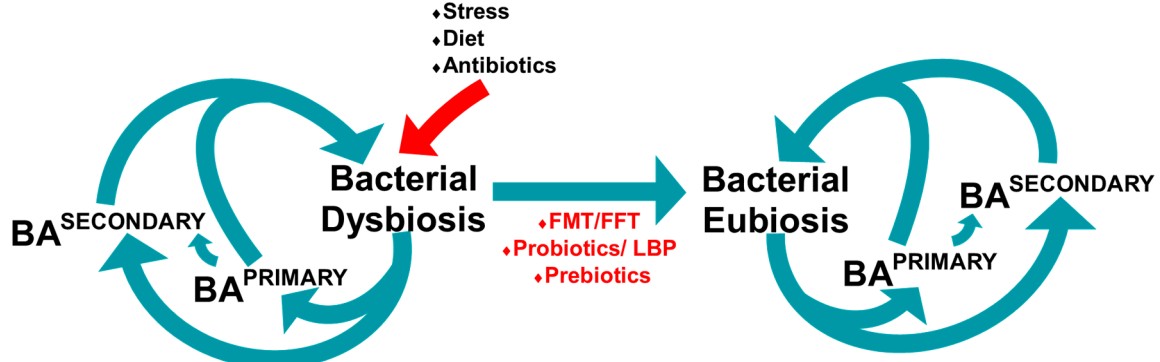

**Figure 4.** Intestinal bacterial homeostasis maintained by primary and secondary bile acids. Stress, diet and antibiotics induced dysbiosis are treated by manipulating the microbiome via faecal microbiota transplantation (FMT); faecal filtrate transfer (FFT); and live biotherapeutic products (LBP).

The cause of primary BAD is unknown. Few studies have investigated the bile acid profile in SeHCAT diagnosed BAD patients. A potential hypothesis for the cause of BAD could be due to changes in the microbiome, as studies demonstrate that patients with BAD have reduced bacterial diversity [28]. Bacterial bile acid transformations shape the bile acids that are available for binding to FXR. Bile acids bind to FXR as both agonist and antagonist with differing affinity and efficacy levels, leading to various levels of the activation and expression of FGF19. This can result in an over-production of primary bile acids and may lead to primary BAD. Establishing the causality of microbiome constituents for health and disease remains a tremendous challenge. In addition, the bacterial species and strains that make up the microbiome are highly specific for the individual; no two individuals are colonised by the same assemblage of microbiota [148]. In order to accurately understand the role of the microbiome in BAD patients, functional genomic analysis of the microbiome of SeHCAT diagnosed patients coupled with metabolic analysis is needed.

Primary BAD is a common GI disorder, estimated to affect 1 in 100 individuals in the UK [4]. Despite this and advances in diagnosing BAD, the current treatment remains flawed, often having undesirable effects [19]. There has been increasing evidence of the promising role and therapeutic potential of the human gut microbiome and its potential for treating BAD. Further studies are needed to link gut microbiota and metabolic features in SeHCAT diagnosed BAD patients to better understand how the microbiome could influence this condition. In addition, further in vitro and in vivo studies would be beneficial in mapping out individual bile acids effects on FXR activation. Finally, clinical trials are necessary to investigate whether FMT, FFT, probiotics and other microbiota altering treatments are a viable therapeutic option to treat SeHCAT diagnosed BAD sufferers.

**Author Contributions:** Writing of manuscript and creation of figures, E.B.M.H.; reading and modifications of manuscript, S.R., D.C., R.P.A., E.M.H.W. and G.C.A.A. All authors have read and agreed to the published version of the manuscript.

**Funding:** This paper is based on independent research commissioned and funded by the NIHR Policy Research Programme (NIBSC Regulatory Science Research Unit).

**Institutional Review Board Statement:** The views expressed in the publication are those of the author(s) and not necessarily those of the NHS, the NIHR, the Department of Health, "arms" length bodies or other government departments.

**Informed Consent Statement:** Not applicable.

**Data Availability Statement:** No new data were created or analysed in this study. Data sharing is not applicable to this article.

**Acknowledgments:** Thank you to all the co-authors that supervise my PhD. Thank you to the microbiome group at NIBSC; Saba Anwar, Chrysi Sergaki and Alastair Logan.

**Conflicts of Interest:** The authors declare no conflict of interest.

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
