# Peer review of "Manipulating the Microbiome: An Alternative Treatment for Bile Acid Diarrhoea"

_2036-7481, doi:10.3390/microbiolres12020023_

Round 1

Reviewer 1 Report

Line 77: It might be helpful for the reader to read about the IBS symptoms and clinical signs -> to be able to distinguish between IBS and BAD.

Line 78 – 81: Literature/Citation?

Line 83: Is this the only change in diet? Are there any suggestions? Especially related to the microbiome?

Line 133-137: The context is not very clear. Are there more BAs in BAD patients than in IBS-D patients? Which BAs?

Line 140: Check the literature! Cited paper says “may contribute to total BA synthesis” -> so it is not a fact. Be careful with citing incorrectly.

Line 204 ff: Why is this section (4.2 -4.3) relevant for this review?

Line 302:-305: What is meant here? Be more careful with your wording, e.g. “it could explain the increase of…”, the science behind it might be much more complex!

Line 331: Consistency -> spell out TGR5 as done before with FXR; where else in the body is TGR5 expressed?

Line 359: “This review focuses on the gut microbiome.” Sentence is quite odd too write on page 11 out of 15.

Line 361-65: Context? Why is this information relevant?

Line 375: rephrase sentence

Line 386 ff: This is again a very simplified explanation – all diseases are due to alterations of only 2 molecules?

Line 391 ff: This section should focus on BAD, however, is mainly about data of IBS-D (which was explained at the beginning to be very different to BAD).

Line 445-447: Citation?

Line 447-448: Citation?

Line 454: So far FMT was unsuccessful in IBS, and this is very important to mention it, especially in a review.

Figure 2: What is meant with BAI? Abbreviation?

Reviewer 2 Report

Research Summary:

The author’s present a review of BAD to include an introduction to clinical and economic concerns around primary BAD, the synthesis, structure and function of bile acids (including secondary BA), loose microbiome ties to BAD and better mouse model ties of gut microbiota to BAD, and general microbiome manipulation information (e.g. probiotics).

General Perception:

The author’s present a mostly well written review. I feel that the flow and impact of the review is inhibited by its current structure and lacks adequate ties in some topics or relies on loose inferences that could be better spelled out for readers. I am missing a call to action or directions for future study out of the review and am left with more of a “ok, this makes sense but now what” impression after reading. Section 7, in particular, requires major rewrites in my opinion. Detailed suggestions are outlined below.

Detailed comments and questions on sections:

General:

  • Double check comma usage throughout.
  • General proofreading throughout.

Figures:

The figures seem appropriate and adequately appealing for the material.

Abstract:

The abstract seems well written and appropriate. I would suggest adding an impact statement at the end of the abstract summarizing the author’s goal in writing this review.

Suggestions:

18-19   Summarize a few of the “undesirable effects” of treatment.

22-23    “…taking antibiotics could contribute…” to “…introducing antibiotics may contribute…”

  1. Introduction:

Suggestions:

46-47   A citation seems appropriate for this claim.

82-92   This feels like a missed opportunity in review with diet changes being well documented in gut microbiome modulation. Here, you explain diet changes as treating symptoms only when it could be treating causal as well if gut microbiota are causal of BAD as later implied.

  1. Structure and function of Bile Acids

This section feels like it should go after “3. The Enterohepatic Circulation and Primary Bile Acid Synthesis” and before “5. Bile Acid Receptors and Downstream Targets.”

  1. The Enterohepatic Circulation and Primary Bile Acid Synthesis

141-142           “there is” to “there are”

148                  Indent to match paragraph structure

  1. Secondary Bile Acids

No major comments here.

  1. Bile Acid Receptors and Downstream Targets

No major comments here

  1. Microbiome

I feel the general information here not relating to BAD is too in-depth and voluminous. I would suggest cutting back significantly on defining what a microbiome is and what the gut microbiome is. Put the focus more on emphasizing what the possible connections to BAD are. I suggest either combining the opening section and 6.1 or doing away with them completely and integrating the relevant information into 6.2 and 6.3. 6.2 and 6.3 are the primary sections of interest here and I might also suggest putting 6.3 before 6.2. In this case you have a mouse model study demonstrating a linkage between gut microbiota and BA. Then you have the human system which is less well studied or defined.

  1. Manipulating the microbiome

Capitalize “microbiome” in the section title to be consistent with other section headers. Again, the information presented in this section is very general and not specifically tied to BAD. The authors should use the reviewed studies to suggest work to be done in BAD. This section is very disconnected from the rest of the review. The authors should suggest means by which FMT or probiotic treatment could directly impact BAD or studies that could be undertaken to explore treatments.

  1. Conclusion

Conclusions feel disconnected from the wealth of information and the story presented in the review. I suggest that the authors rewrite this section to summarize the story from the review, offer a unifying theory, and either make a call to action or suggest future lines of research supported by the review.

Reviewer 3 Report

This is a scientifically sound paper reviewing current knowledge on the microbiome and its role in transforming primary bile acids into secondary bile acids. Therefore, in the case of microbiota shifting and dysbiosis due to diet patterns, stress, medicines, antibiotics primary BAD( Bile acid diarrhoea) occurs. In this case, therapeutical approaches through microflora manipulation could be beneficial for the host.

It is an interesting paper which could contribute to obtaining global knowledge in the field. The paper is a well documented and written  . It should be of high interest to the scientific community , specifically for those working in this scientific area.

The paper could be published in its present as it is well written , and included bibliography is up to date 

Round 2

Reviewer 1 Report

No comments

Author Response

This manuscript is a resubmission of an earlier submission. The following is a list of the peer review reports and author responses from that submission.

Round 1

Reviewer 1 Report

This article by Evette B.M. Hillman et al. described very interesting data on Bile Acid Diarrhoea. Even if this article is well-written and interesting, my opinion is that this article would get a better audience in a mdpi journal specialized in biochemistry, as most of the content of this manuscript focuses on the bile acid modifications and less on the microbiome.

In part 2: a Flow diagram to understand the cascade of reaction would be very usefull.

Global: numbers under thirteen must be written in full letters

Line 340- 341: these recommendations are not appropriate, as this manuscript is not focused on this general topic.

Line 365 to 370: this part has to be detailed as this data could be important to understand.

Line 404: These observations are mostly due to bacteriophages. The authors have to discuss this.

Reviewer 2 Report

Hillman et al. submitted their manuscript entitled „ Manipulating the Microbiome: an Alternative Treatment for Bile Acid Diarrhoea“ to Microorganisms. The title of the manuscript and chosen MDPI journal indicate that the primary attention will be paid to the microbiome. However, the main attention is paid to the metabolism of bile acids, but their relation to the microbiome is superficially described only. Moreover, a part of the modified metabolism/conversions of the bile acids in the absence of microbiome was completely omitted. I mean the bile acid metabolism in germ-free animals.

A have other objections to the manuscript:

L29: Watery diarrhea - It will be interesting to include changes in tight junction proteins (intestinal barrier) of the host during diarrhea.

L340: You sent the manuscript to Microorganisms. You paid attention to the physiology/biochemistry of bile acids. Now you had the occasion to pay attention to the microbiome, but you refused it? Why did you not chose a biomedically oriented journal for submission of your manuscript?

L341: "we suggest" - I think that it is not the best way to point on some reviews. I recommend modifying the sentence, e.g., "we suggest reviews by Author 1 et al. [101], Author 2 etal., [102] and Author 3 et al. [103]. The same was done on line 344. 

L351: See Sagar et al. comment.

L351: C. leptum C. should not be abbreviated because it is used for the first time.

L384: El-Salhy et al. - see Sagar et al.

Round 2

Reviewer 1 Report

The authors have completed most of my advices/comments, so I could recommend publication of the manuscript, after minor revisions.

  • I maintain nevertheless my recommendation to target another more biochemistry-specialized journal, as the readers would risk to be disappointed, regarding to the little part focusing on microorganisms.
  • A scheme summarizing the conversion of primary bile acids into secondary bile acids would be useful, at least to sumarrized visually the numerous reactions described in the manuscript.

Reviewer 2 Report

Dear Authors,

Many thanks for your comments and explanations. I very evaluate your effort to improve your manuscript. However, you added texts that are irrelevant to the reviewed topic (L347-351 and 356-396). Moreover, the ratio of 1 (host) to 10 (microbiome) cells has been overcome (Sender et al., Plos Biol, 2016 and Sender et al., Cell, 2016), but you can have a different opinion on these counts, of course. The proportion between the biochemistry of the bile acids and microbiome-related information remains inadequate. I have believed that another MDPI journal would be a better choice than Microorganisms.

L80: track versus tract?